# Polymer Science and Engineering Using Deep Eutectic Solvents

**DOI:** 10.3390/polym11050912

**Published:** 2019-05-21

**Authors:** Ana Roda, Ana A. Matias, Alexandre Paiva, Ana Rita C. Duarte

**Affiliations:** 1LAQV, REQUIMTE, Departamento de Química da Faculdade de Ciências e Tecnologia, Universidade Nova de Lisboa, 2829-516 Caparica, Portugal; ana.roda@ibet.pt (A.R.); alexandre.paiva@fct.unl.pt (A.P.); 2iBET, Instituto de Biologia Experimental e Tecnológica, Apartado 12, 2781-901 Oeiras, Portugal; amatias@ibet.pt

**Keywords:** deep eutectic solvents, polymers, green chemistry, synthesis, extraction, formulation

## Abstract

The green and versatile character of deep eutectic solvents (DES) has turned them into significant tools in the development of green and sustainable technologies. For this purpose, their use in polymeric applications has been growing and expanding to new areas of development. The present review aims to summarize the progress in the field of DES applied to polymer science and engineering. It comprises fundamentals studies involving DES and polymers, recent applications of DES in polymer synthesis, extraction and modification, and the early developments on the formulation of DES–polymer products. The combination of DES and polymers is highly promising in the development of new and ‘greener’ materials. Still, there is plenty of room for future research in this field.

## 1. Introduction

Deep eutectic solvents were first described by Abbot in 2002 [1], referring to the liquid formed just by mixing quaternary ammonium salts with metal salts, at ambient temperature. The term ‘eutectic’ was first introduced at the end of the 19th century (1884). By definition, it refers to the minimum freezing/melting temperature achieved by a mixture of two or more compounds at a particular molar ratio [2]. However, the deep eutectic solvents (DES) concept is often not restricted to a unique ratio of compounds, being commonly defined as a mixture of two or more compounds with a lower freezing/melting point lower than the pure components. This concept opens a larger spectrum of DES options with tunable properties [3]. The mechanism of DES formation is not yet well understood but it most probably occurs through hydrogen bonding, electrostatic interactions and/or Van der Waals interactions [4,5,6,7,8]. Since there is no consensual definition for DES in the scientific community, all the papers reporting the use of DES, under their own definition, were considered in this review.

The exponential concern with environmental issues highly motivated the development of green technologies. DES have emerged as green solvents due to their low toxicity and volatility, biodegradability, biocompatibility, easy production, high yields and purity, and the fact that they are widely available precursors [9,10]. This type of solvent can fulfill the principles of green chemistry [11,12] while being highly tunable for several distinct areas. Good examples of DES that “fully represent green chemistry principles” are natural DES (NADES), a subclass of DES formed by compounds from natural origin [13,14,15].

Throughout the last decade, an exponential interest has been given to the research on DES, a rate expected to maintain if not to increase (Figure 1a). Up until now, deep eutectic solvents have already been reported for several applications, primarily for electrochemistry, synthesis and extraction [14,16,17].

Within all the reported studies, ~8% correspond to the use of DES in polymeric applications [18]. The investigation on this field largely increased in the recent years (Figure 1b), including solubilization, extraction, synthesis, or modification of polymers. The most recent studies report DES incorporation in the formulation of polymeric products. The current relative incidence of the different areas herein explored is represented in Figure 2.

Still, it seems that the DES versatility regarding polymers seems to be broadening. This review summarizes the explored applications of DES and polymers, aiming to consolidate current achievements and to promote further developments.

## 2. Fundamental Studies Involving DES and Polymers

The investigation of deep eutectic solvents for several applications has been growing at a fast pace. The fundamental knowledge of their properties for those applications is hence crucial for their optimization and development. Although DES are frequently fully characterized as individual systems, the intra- and inter-interactions in the application context are essential. The fundamental studies involving DES and polymers are scarce, as expected of novel applications, hence we present here studies reported within this topic in the last 3 years.

In 2016, Sapir et al. studied the molecular solvation of poly(vinyl pyrrolidone) (PVP) polymer in choline chloride:urea 1:2 (molar ratio) DES, in comparison to water, through conformational and thermodynamic analysis. Despite the similarity of DES and water as solvents for PVP, the authors reported different intermolecular interactions between PVP-DES and PVP–water. According to Flory−Huggins Solution Theory, the PVP individual interactions are more similar to the DES’ than water interactions, favoring PVP-DES interaction. Their results supported this DES as a “close-to-ideal” solvent for PVP [19]. Similar studies were carried out for poly(ethylene oxide) (PEO) polymer in 1:2 (molar ratio) DES of ethylammonium bromide (EABr):glycerol, EABr:ethylene glycol, or butylammonium bromide (BABr):glycerol. The results characterize the tested DES as moderately good solvents for PEO with higher quality for ethylene glycol than glycerol-based DES [20]. Moreover, they postulate that the higher density of hydrogen bonds per unit volume and the availability of such interactions improves the DES solvation capacity. Fundamental data on the influence of the DES hydrogen networking in its 3D structure can be acquired to predict its solvation capacity and used as a preselection parameter for certain applications. An example is the work of Hammond et al. on neutron diffraction and atomistic modelling to acquire the probable liquid structure of choline chloride–urea DES [21]. Considering DES–polymer systems, this type of preliminary information for several DES systems and/or different component ratios could be used to select or adjust suitable DES for polymer solvation. However, the DES–polymer systems are more complex than DES in its isolated form. It is then necessary to study the combined system to have more accurate information on their interaction. In this context, Stefanovic and his coworkers performed a systematic quantum chemical investigation on the PEO solvation in DES of choline chloride (ChCl) mixed with urea, ethylene glycol, or glycerol. The solvation conformation of PEO was correlated with the type of hydrogen bond donor, the density of hydrogen bonds in DES and the influence of PEO in the hydrogen network. ChCl:urea had the strongest and denser hydrogen network, whereas ChCl:glycerol presented the weakest and less dense. Interestingly, the PEO ChCl:urea disrupted the weaker hydrogen bonds, strengthening the robust ones, which caused a highly structured solvation environment, ‘imprisoning’ the PEO in a static and coiled conformation. On the contrary, in the least structured environment of ChCl:glycerol, the polymer presented a free conformational structure [22].

A different study, conducted by Hillman et al. observed the intrinsic composition of DES and the extrinsic coupled conditions needed for electrochemical deposition of polyaniline (PANI) in aqueous media and DES of ethylene glycol:choline chloride or oxalic acid:choline chloride. Furthermore, the influence of DES as electrolytes in the electroactivity of the produced PANI-based films was studied and evaluated in terms of film longevity and charge storage stability after successive redox cycles. The oxalic acid-based DES electrolyte was better for PANI electrodeposition, with no further additives, whereas ethylene glycol-based DES required the addition of sulfuric acid as a protic source [23].

Thermodynamic investigation of phase equilibrium involving DES:polymer:water systems was carried out by Baghlani and Sadeghi. DES of choline chloride with urea, ethylene glycol or glycerol 1:2 molar ratio and the water-soluble polymers polypropylene glycol400 (PPG400), polyethylene glycol400 (PEG400), and polyethylene glycol10000 (PEG10000) were used to prepare different DES/polymer/water ternary systems. Their water activity was determined through the isopiestic method. The authors observed that the DES:PPG:H_2_O mixture formed two immiscible aqueous phases, called aqueous biphasic systems (ABS), due to the incompatibility of both polymer and DES to form hydration complexes. Specifically, the choline chloride and glycerol caused soluting-out of the polymer. This mechanism was depended on the competition between hydrogen bonding and hydration. Similar studies on ABS have been reported by Freire et al. for ionic liquids, also contemplating polymer-based ABS [24].

## 3. Polymeric Synthesis Using DES

The first article identified reporting polymerization using an eutectic mixture was published in 1985 [25], 100 years after Guthrie published the eutectic definition [2]. In that study, Genies and Tsintavis compared the preparation of the polyaniline polymer by electrochemical polymerization in a eutectic mixture of NH_4_F-HF to other media (aqueous and organic). By that time, they could already identify the eutectic solvent as an advantage over the use of other solvents. The polyaniline polymers produced in the eutectic mixture presented better nucleation and polymerization process, with almost 100% yield, as opposed to other media. Additionally, it was also better in terms of adherence and electrochemical properties for all the preparation conditions studied. Over the years, other studies were conducted on the synthesis of natural and synthetic polymers evolving eutectic mixtures [26,27,28], but only in 2011 the term ‘deep eutectic solvent’ was used by Mota-Morales et al. for polymerization studies [29].

### 3.1. DES as Functional Monomers

In the study of Mota-Morales et al., the authors used mixtures of choline chloride with acrylic acid or methacrylic acid monomers to form DES, providing both media and functional monomer for frontal polymerization. From their knowledge in similar works using ionic liquids, they could tailor the ratio of DES components to enhance polymer conversion through a suitable content of double bonds and by stabilizing the reaction temperature and velocity through the high viscosities of the prepared DES. Moreover, once the monomer units contained in the DES were polymerized, it was possible to reutilize the choline chloride component. The mentioned process allowed the utilization of the DES as a medium and functional monomer for polymerization while demonstrating an enhanced performance in comparison to conventional organic solvents and ionic liquids [29]. DES enhanced the polymerization process at four different levels: (i) through the reduction of components for reaction, (ii) by acting as an alternative solution to organic solvents with improved performance, (iii) minimization of the waste produced, and (iv) recycling of the remnant compounds. These features highly enrich the “green” value of DES.

Additional studies about the use of DES as a functional monomer for polymerization were published. The groups of Xu, Wang J. and Wang R. reported the polymerization of choline chloride:itaconic acid (ChCl:IA), using IA as a monomer unit, for the preparation of solid extraction matrices [30,31] or stationary phase for chromatographic separation [32]. Xu et al. noticed unique benefits of DES polymer incorporation in the extraction system in terms of its stability, surface area, and moldable structure [30]. In turn, Wang J. et al., mentioned the potential of different monomer compositions in DES to produce tuned-sorbents for extraction [31]. In another work, Isik et al. produced poly(ionic liquids) for CO_2_ sorption by photopolymerization of DES monomer units formed by 2-cholinium methacrylate bromide monomer and natural carboxylic acids, amidoximes, or amine [33]. A more recent paper described a novel DES of 3-acrylamidopropyl trimethylammonium chloride (APTMACl):d-sorbitol 2:1 (molar ratio) as a monomer for polymerization on the surface of amino magnetic metal–organic framework (Fe_3_O_4_-NH_2_@HKUST-1-MOF). This DES was designed for the functionalization of Fe_3_O_4_-NH_2_@HKUST-1, not only through polymerization but also by coordination of DES-nitrogen and HKUST-1. The functionalization with DES incorporated hydrogen and ionic interactions on the surface of the MOF sorbent, which improved its function in the extraction of cationic dyes [34].

### 3.2. Electrochemical Polymerization

Electropolymerization was also mentioned in Mota-Morales and coworkers’ review. It consists in the polymerization of monomers through electrochemical induction that leads to the formation of conductive polymers, normally in the form of electrodeposited films [35]. An interesting feature of using DES as media for electropolymerization is its influence in the polymer formation and its final characteristics [36,37]. This type of polymerization is one of the first applications of DES, reported in the 1980s, as stated by Fernandes et al. in their article about polyaniline (PANI) electrosynthesis in DES. PANI electropolymerization rates in DES of choline chloride with ethylene glycol, urea or glycerol showed to be determined by the viscosity and conductivity of those solvents [38]. Using the same DES systems, Prathish et al. published the electropolymerization of poly(3,4-ethylenedioxythiophene) (PEDOT), with better results of stability and sensitivity for the PEDOT prepared from choline chloride:urea DES. The shape, size, surface, electrocatalytic, and sensing properties of the PEDOT films were highly influenced by the DES used [37,39]. The most recent study reported on this topic, from Zou and Huang, also focused the electrodeposition of PANI from DES. The novelty of their work is the electropolymerization without exogenous acid as a proton source, by using newly synthesized DES, of proton-functionalized anilinium hydrochloride ([HANI]Cl) or anilinium nitrate ([HANI]NO_3_) with glycol. The authors tailored the ratio of anilinium salt:glycol in terms of conductivity and obtained a final DES with high conductivity and low viscosity, highly suitable for electropolymerization [40]. Table 1 resumes the polymers synthesized through electropolymerization, the DES used for that purpose, their function, and the potential applications of the produced products.

### 3.3. Polycondensation

Polycondensation occurs in polymer synthesis by condensation reactions through molecular combination with liberation of small molecules (byproduct), normally forming polymeric matrices [43]. This type of polymer synthesis is commonly used for the production of carbon-based materials. The review of Carriazo et al. from 2012, refer some polycondensation studies with DES [44], however the investigation on this topic is still scarce. In the last 5 years, three representative studies of polycondensation in DES were published [45,46,47]. Lopéz-Salas et al. reported resorcinol:urea (3.5:1 molar) and resorcinol:urea:choline chloride (3.5:0.5:1, 3.5:1:1, and 3.5:2:1 molar, respectively) DES for tailoring the structure of hierarchical porous carbons. In their study, the urea content in DES allowed controlling the pore dimensions of the produced matrices [47]. Patiño et al. used DES of resorcinol:3-hydroxypyridine (1:2:1 molar) or resorcinol:tetraethylammonium bromide (1:3:1.75 molar) as polycondensation solvent to produce hierarchical nitrogen-doped carbon molecular sieves [45]. Chen et al. published the use of 1:2 (molar ratio) choline chloride:urea DES as a solvent for the two-stage polymerization from phenol and formaldehyde, to produce porous carbon xerogels [46]. Interestingly, in both Chen and Patiño works, DES act as a homogenization medium, structure-directing agent and as a nitrogen source for carbons [45,46].

### 3.4. Molecular Imprinted Polymers with DES

Molecular imprinted polymers (MIPs) are molds made by polymerization around template molecules, for its specific recognition. By molding specific structures and size in the polymeric matrix, the MIPs are able to specifically recognize and hold the molded template by complementary match [48,49]. These polymer molds are used for recognition and separation, purification, production of artificial antibodies, target delivery and/or electrochemical sensors [48].

One of the most recent applications of DES is its use for the production of MIPs, either as medium or solvent [50,51], as MIPs modifier [51,52,53,54,55,56,57,58,59,60,61,62,63], as MIPs functional monomer [48,53,54,64,65,66,67,68,69,70,71,72], and even as MIPs template [67,69] (Table 2). From the whole publications involving DES and polymers, more than 15% correspond to DES and MIPs, reported just in the last four years [18]. In 2016, Li et al. described for the first time the use of DES for the modification of MIP. The authors postulated that the interaction of DES with the functional monomer, and/or the surface of MIP, improved affinity, selectivity and adsorption of chlorogenic acid [53]. In the same year, Liu et al. reported the use of DES as a functional monomer for MIPs polymerization: DES-MIP. The produced DES-MIPs have shown to be stable, reusable, have a high imprinting factor, fast binding kinetics, and high adsorption capacity. The authors described the produced DES-MIPs as ‘outstanding’ for the specific and selective recognition of bovine hemoglobin from protein mixtures or real samples [64]. All the following works showed similar features, supporting DES-MIPs as highly efficient and selective in the recognition of the template molecules. Some authors also reported the advantages of the produced DES-MIPs in comparison with MIPs from conventional monomers [52,53,56,66]. According to Li G. et al., Li X. et al. (2017), and Xu et al., DES have an advantage over conventional monomers due to their high content in available functional groups, allowing unique interactions with the template molecules, which translates into higher affinity and selectivity for the respective DES-MIPs [48,52,53,54,57,62]. Li G. and its coworkers also postulated that increasing DES-MIPs rigidity can prevent its shrinkage or swelling [53,54,55,56]. Moreover, the liquid character of DES may also be an advantage by including the monomer in the bulk of DES or by substituting the need of media or solvent when used as an additive.

The most recent paper reported on DES-MIPs showed some innovative features. Fu and coworkers prepared DES-MIPs in a 2D material as a surface, using a 1:2 molar ratio mixture of choline chloride and acrylic acid DES, as functional monomer. The use of this technique avoids the immersion of the template during polymerization, facilitating its removal. The produced polymeric matrix presented the required biomolecule recognition and showed to be renewable. Additionally, it presented antibacterial activity, a high added value feature for analyzing or delivering additives to biosamples while preserving them [72].

### 3.5. Other Types of Polymer Synthesis

Free radical polymerization is one of the most common types of polymerization. Mota-Morales et al. (2017) published a very complete review of polymerization using DES focusing on the ‘free radical polymerization’ of and in ‘deep eutectic solvents’. Herein, we make a brief summary of the main outcomes mentioned in their work and recommend the consultation of their review as a complement to this section. It states the high plasticity of DES, either in terms of composition, molar ratios and the broad range of interactions. The authors describe the use of DES as inert solvent and/or reactive component, focusing the ‘all-in-one’ systems when both functions are combined. They summarize the polymerizable DES, the polymer resulted from their free radical polymerization and their application. Their review highlights the green character of DES, its ‘greening’ impact in the polymerization process and in the product formed. In addition, it presented the potential of DES as a broadener of the synthetic conditions (e.g., temperature and vacuum) in comparison to other solvents, creating new polymerization strategies [74]. After this comprehensive review, innovative work has been developed in the field.

Ferreira et al. synthesized chondroitin sulfate mesoporous materials by using DES for biopolymer dissolution and mesoporous templating. Despite the lack of knowledge about the exact templating mechanism, the authors suggested that DES can act as a capping agent and/or as filler. Mesoporosity was effectively obtained by including DES in this biopolymer composite synthesis, while a stable structure was maintained after DES removal [75].

In turn, Maximiano and coworkers published the first article reporting the use of DES as cosolvent in ethanol for supplemental activator and reducing agent atom transfer radical polymerization (SARA ATPR) [76]. Following this new development, Wang et al. and Mendonça et al. published the first studies on the use of different DES as ligand [77], as 100% solvent for SARA ATPR [78] or both [77]. Other studies also reported the use of DES as solvent medium for polymer synthesis [46,79,80,81,82,83,84,85,86,87,88,89,90]. In particular, the communication of Park and Lee on the polymerization of 3-octylthiophene in DES presented interesting features regarding the use of DES as a solvent in polymerization. DES of choline chloride:urea improved polymerization yield and duration in comparison to chloroform and the best ionic liquids for that purpose. The hydrogen bond basicity of the DES used showed to be preeminent for the effective polymerization of that polymer [81]. Furthermore, a more recent work published the use of ChCl:glycerol 1:2 molar ratio DES as reaction media for oleofin anionic polymerization [90]. The authors highlighted the advantages of the protic and polar character of DES for the polymerization and reported that the selective use of glycerol as hydrogen bond donor instead of urea, lactic acid, or oxalic acid increased the polymerization performance. These examples emphasize, once again, the potential of DES versatility to be tailored for specific applications. The substitution of organic or other toxic solvents by green DES and its role in additional functions, simultaneously to solvent, is a great development in green chemistry.

## 4. Extraction of Polymers with DES

Within the DES applications already reported, its use for the extraction of polymers has been widely explored. A recent review of Zdanowickz et al. (2018) reviewed the work developed regarding DES for extraction of polysaccharides, namely, lignin, cellulose, starch, agar, agarose, chitin, chitosan, xylan, pectin, and inulin. It summarizes the DES used for the extraction of these polymers from different biomasses, either for separation from the raw materials or for polysaccharides recovery/purification, through solubilization or induction of fibrillation or crystallization. It also addresses the use of DES for obtaining low molecular products from polymers degradation or saccharification [4]. Moreover, several reviews address polymer extraction with DES from lignocellulosic biomass, one of the most explored raw materials regarding processing with DES [91,92,93].

As a result of the intense investigation on the topic, the discoveries are steadily uncovered. In the present review, recently published works involving DES in polymer extraction are listed in Table 3, summarizing the DES composition, the polymers extracted, the raw material used for extraction and some additional information about the cited papers. Some previously reported studies on this topic are available in the review of Zdanowickz et al. [4].

In a paper of Saravana et al., they tested 14 different DES for chitin extraction from shrimp waste. All the DES presented higher yields than the conventional extraction method and ChCl:malonic acid was the more selective DES for pure chitin. Additionally, films from the extracted chitin were efficiently produced, with similar characteristics to films from commercial chitin [94].

In terms of the parameters influencing polymer extraction using DES, Bai et al., and Mammilla et al. reported the DES acidity, viscosity, and the content in free hydrogen proton as main features influencing their extraction efficiency [95,96]. Moreover, the solubility of the target polymer in the DES is obviously crucial for the extraction efficiency. The reviews of Melro et al. and Duan et al. summarized DES found to be suitable for the dissolution of lignin [97] and chitin [98], the most commonly extracted polymers with DES. Additional features regarding the DES mechanisms for extracting polymers are normally not referred, probably by assuming to be basic extraction principals. Herein, we present some of those basic concepts, focusing polymer extraction with DES and infer some unique possible contributions of the DES for this application. The diffusion of DES into the matrix containing the extractable polymer and the mass transfer interactions of DES–matrix–polymer are decisive factors. The ionic content and hydrogen bonding capacity of DES is highly promising for mass transfer improvement and for rupturing conformational chemical bonds of the matrix, facilitating solvent diffusion. Furthermore, as referred in the previous sections for other applications, DES can establish hydrogen and electrostatic interactions with the functional groups of a target molecule, conferring them higher affinity to each other. This principle is applicable to polymers extraction with DES. The knowledge of the molecular structures involved in the extraction and their possible interactions can be used to tailor the extraction yield and/or extraction selectivity of target polymer(s).

## 5. DES as Polymer Modification Agents

In the previous sections, the use of DES was mentioned as a modifier agent, for example, in the synthesis of DES-MIPs as porogen or by incorporation of DES properties into the MIPs, conferring unique interactions to recognize target compounds. These unique interactions can occur through derivatization of the polymeric structures, either by derivatizing their own functional groups or by incorporating DES in the structure.

### 5.1. Derivatization

Several studies were conducted on the use of DES as a derivatization agent by its inclusion in a polymeric material. We present here some examples. Wang and the coworkers reported the incorporation of DES into a polymer monolithic cartridge (DES-M), which could improve the affinity and selectivity of the solid-phase extraction of quercetin from *Ginkgo biloba*. The observed advantages were attributed to the strong hydrogen bond network of the DES as a source of electrostatic and ionic interactions introduced in the surface of the DES-M during polymerization [103]. Gan et al. also introduced DES (ChCl:glycerol 1:2 molar) as a modifier in the polymeric synthesis of anionic-exchange resins, and as porogen and derivatization agent by its inclusion into the resin [104]. Moreover, Li and Row incorporated aqueous DES of betaine mixed with glycerol, glucose, ethylene glycol or urea into mesoporous materials, improving their efficiency for dextrans separation [105].

Other types of polymer derivatization using DES were published. Bangde et al. used ChCl:urea and ChCl:glycerol (1:2 molar) as chitosan methylation solvent. The urea-based DES was selective for N-methylation while glycine–DES systems also produced O-methylated chitosan. Both systems mediated effectively chitosan methylation and presented advantages over the conventional method: (i) reduced organic solvent, (ii) decreased reaction time, and (iii) no polymer scission [106]. A similar but innovative study reported O-acylation of chitin, directly converted from shrimp shells by DES. The wide versatility of these solvents, allowed for their simultaneous use for demineralization, deproteinization, and acylation initiator, while preventing the use of acids, bases, catalysts, and other acylating agents needed for the conventional method. Choline chloride:_DL_-malic acid 1:2 (molar ratio) were considered optimal for this purpose, within other ChCl:malic acid and ChCl:lactic acid DES systems [107].

Cationic derivatization of polymers using DES has also been described [108,109]. Vuoti et al. used a boric acid:glycidyl trimethylammonium chloride 1:3 (molar ratio) DES mixture for the enhanced cationization of cellulose biopolymer, aimed wastewater treatment. The cationic cellulose derivatized by DES was effectively used for water treatment, being a potential competitor to commercial polyacrylamides [109].

### 5.2. Plasticization

DES have also been used as plasticizer of polymers. The review of Wong et al. comprises some general characteristics of plasticizers and their influence in polymers properties [110], listed in Table 4.

According to these principals, deep eutectic solvents have suitable properties to be used as a plasticizer. Table 5 compiles some DES used for the discussed purpose, the plasticized polymers, and the properties conferred. Those properties can change for different polymers [111,112], and even for the same polymer with variations in structure or MW [113]. In sum, the influence of DES in polymer plasticization depends on several factors, including DES composition, polymer type, DES:polymer ratio [114] and, of course, their specific interactions.

An additional interesting feature also reported by Wong et al. is the possibility to use internal or external plasticizers, where the internal type usually implies the introduction of monomers into the polymeric matrix [110]. Within the studies of DES as a plasticizing agent, we could not find DES as a monomer for polymer plasticization. Since the use of DES as a functional monomer was already reported for the synthesis of polymeric structures, we would like to highlight its potential to be explored as monomer plasticizer.

Still, within the context of the use of DES for polymer plasticization, Abbott et al. studied DES of ChCl:glycerol, ChCl:ethylene glycol, and ChCl:urea as plasticizers of high-density polyethylene (HDPE). Despite the improvement of polymer ductility after modification with DES, its strength and *T*_g_ were not significantly changed. For this reason, the authors suggested that the modification of DES in the HDPE was as a lubricant rather than a plasticizer, reveling an additional modifier function of DES. More interestingly, the DES modified-HDPE was blended with starch plasticized with DES. DES-modifications allowed to blend, for the first time, polyolefins (HDPE) to nonchemically modified carbohydrates (starch) [115].

The thermal and mechanical polymer properties conferred by plasticization with DES can attribute or influence other properties, specific of certain applications, as polymer foaming enhancement for the production of 3D porous materials [116] or water retention capacity, for the production of impermeabilized or hydrophilic materials [117].

### 5.3. Other Modifications

Some DES can transform a polymer’s structure by disrupting or rearranging its chemical and/or physical networks. Within this context, acidic DES have been used for protein denaturation [107,122]. Tan and the coauthors published the use of urea:guanidine hydrochloride DES as protein denaturant to transform silk fibers into nanofibers [122]. The mentioned study used DES for the combined transformation of two different biopolymers, using protein denaturation with DES as a strategy for the modification of a more complex biopolymer matrix.

Deep eutectic solvents can also be used as reactants for polymer modification. An example of this type of modification was published by Lian et al., which used zinc chloride:urea 3:10 (molar ratio) DES as both solvent and reactant for lignin modification. The zinc contained in DES was partially integrated into the lignin structure by chelation with the lignin functional groups, conferring to the modified lignin a 4-fold increased molecular weight and enhanced thermal stability [123]. Crosslinking is another possible application of DES as a reactant, as reported by Jordan et al., which used a deep eutectic mixture of hexaketocyclohexane octahydrate:urea to crosslink chitosan [124].

## 6. Formulation of DES Materials Using Polymers

In the previous sections, DES were explored mainly for polymer processing, as vehicles, additives, modifiers, or just as solvents to achieve certain polymeric products or properties. In this section, deep eutectic solvents are the principal agents of the envisioned product, while polymeric systems are used as modifiers or carriers.

For this purpose, it is important to introduce the concept of therapeutic deep eutectic solvents (THEDES). By definition, THEDES are deep eutectic solvents for which at least one of the mixture components is an active pharmaceutical ingredient [125]. However, bioactive compounds solubilized in DES are also commonly considered as THEDES [126].

Tuntarawongsa and Phaechamud were the first authors to describe the use of polymeric eutectic systems for controlled delivery of active ingredients [127,128]. They used a mixture of two therapeutic compounds—menthol and camphor—to form a therapeutic DES and designed two different delivery systems. The addition of eudragit^®^ polymers to the eutectic liquid up to 40% w/w, increased its viscosity, allowing to form a gel product with potential for topic application, using as an advantage the menthol ability to increase skin penetration [127]. Moreover, an injectable formulation was developed. The same DES system incorporating 30% w/w of eudragit^®^ was used for the solubilization and delivery of an additional active ingredient, ibuprofen. The hydrophobic properties of the DES prolonged the time of drug delivery in comparison to a pharmaceutical solvent commonly used for drug solubilization. Additionally, the viscosity enhancement conferred by polymer addition contributed to even higher delivery retardation [128]. The versatility of these polymeric eutectic systems has a high potential to design delivery systems with tailored sustained drug release.

A distinct approach was presented by Aroso et al. who developed polymeric delivery systems by impregnating THEDES of menthol:ibuprofen (3:1 molar) in a starch:poly-ε-caprolactone polymer blend. The produced 3D structures had higher porous and faster ibuprofen release than the polymeric matrix produced with ibuprofen powder [129].

Furthermore, Mano et al. were the first reporting the encapsulation of THEDES in polymeric fibers by electrospinning [9,130]. In 2015, the authors published the encapsulation of ChCl:citric acid (1:1 molar) DES in poly(vinyl alcohol) fibers [9] and in 2017, the encapsulation of ChCl:mandelic acid (1:2 molar) in gelatin fibers was described [130].

More recently, Liu et al. reported the use of mannose-dimethylurea-water 2:5:5 (molar) as a loading system of lipophilic (curcumin) molecules into a hydrophilic chitosan:alginate hydrogel. The amphiphilic nature of the produced DES could efficiently entrap curcumin in its hydrophobic core while being easily incorporated in the hydrogel due to the hydrophilic surrounding. Moreover, the DES external hydrophilic character promoted its spontaneous diffusion to the water phase during washing, maintaining the curcumin encapsulated in the hydrogel. The formulation of this system aimed to mimic naturally occurring delivery systems from natural DES (NADES) to biopolymeric matrices, like plants (mimicked by the hydrogel). The authors confirmed similar behavior for DES and *Schisandra chinensis* fruit extract regarding the transfer of lipophilic molecules to the hydrogel, which supports the prediction that nature has unique delivery systems, by combining NADES and biohydrogels [131]. Despite that the DES have been removed through water washing in their study, it is possible to formulate identical systems without DES elimination from the polymeric matrix, which might be potentially useful for THEDES delivery in aqueous based systems.

The last study on polymeric delivery of DES was published by Silva et al., which designed fatty acid antibacterial DES with a tuned melting point for an efficient and stable load into a commercial polymeric gauze and easy diffusion from the gauze by melting at physiologic temperature. The tailored DES (lauric acid:myristic acid 1:1 molar) was efficiently loaded into the gauze through supercritical dispersion, being highly potential for wound healing applications [132].

Although the delivery of THEDES from polymeric matrices is highly promising, more DES–polymeric products integrating DES as the principal constituent can be developed. A completely different system is now presented. Qin et al. developed a DES gel supported with gelatin biopolymer for ionic skin applications. The gel produced from ChCl:ethylene glycol 1:2, with 22% gelatin presented a higher stretchability and toughness than the gels formed from the isolated compounds (ChCl or ethylene glycol), even higher than a conventional hydrogel. The incorporation of pressure and strain sensors into the flexible ionic conductive DES gel allowed accurate monitoring of human finger bending and multitouch stimuli. Moreover, its nonvolatile character is an advantage over common hydrogels for a long-term duration [133]. This study opens a new platform for the development of nonvolatile sensors based on DES materials.

## 7. Conclusions

Deep eutectic solvents have shown several functionalities and advantages when used in the field of polymers. From their properties as green solvent (reusable, nontoxic, etc.), to their use as functional components for polymer synthesis or modification, and finally their use as the main or active component of polymer-based formulations DESs are extremely versatile. The countless possible combinations, ratios, and properties achievable can be tailored to specific applications.

In this review, an overview of the developments involving DES and polymers was presented. Our goal was to inform and inspire the scientific community to use, adapt and contribute to the available knowledge and to trigger the progress in the field.

## Figures and Tables

**Figure 1 polymers-11-00912-f001:**
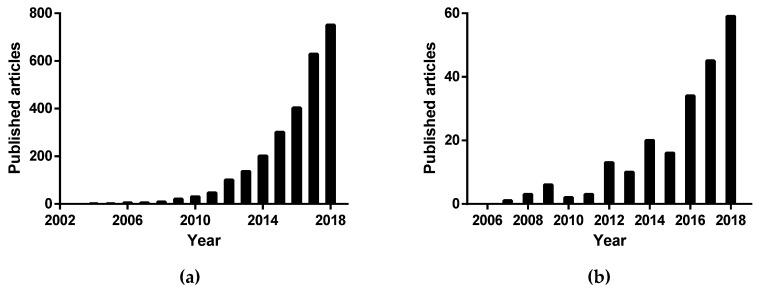
Number of published articles per year on (**a**) deep eutectic solvents and (**b**) deep eutectic solvents and polymers. Data from Web of Knowledge [18].

**Figure 2 polymers-11-00912-f002:**
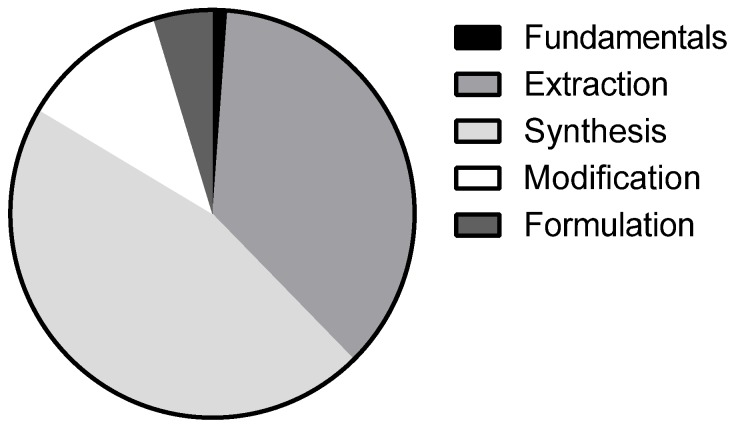
Graphical representation of the relative incidence of the subareas explored within deep eutectic solvents (DES)–polymer field [18].

**Table 1 polymers-11-00912-t001:** Summary of DES and their function in electropolymerization, polymers produced, and their potential applications.

DES (molar ratio)	Function	Polymer synthesized	Application	Ref.
ChCl:ethylene glycol 1:2 ChCl:urea 1:2 ChCl:glycerol 1:2	Solvent Electro-modifier	Polyaniline	Electrochromic devices, supercapacitors	[38]
ChCl:ethylene glycol 1:2 ChCl:urea 1:2 ChCl:glycerol 1:2	Solvent Influence in PEDOT properties	Poly(3,4-ethylenedioxythiophene) (PEDOT)	Sensing of biomarkers	[37,39]
ChCl:ethylene glycol 1:2	Solvent	Poly(methylene blue)	Electrochemical sensors biomarkers	[41,42]
Proton-functionalized anilinium hydrochloride ([HANI]Cl) or anilinium nitrate ([HANI]NO_3_) with glycol 1:10	Solvent	Polyaniline	Capacitors	[40]

**Table 2 polymers-11-00912-t002:** Summary of the work reported in the literature regarding DES-molecular imprinted polymers (MIPs).

DES (molar ratio)	Function	Application	Description	Ref.
ChCl:glycerol 1:2	MIPs modifier: interaction with a functional monomer	Selective recognition and separation of chlorogenic acid from honeysuckle	DES-MIPs are more rigid, preventing shrinking or swelling; DES-MIP showed higher adsorption capacity than MIP	[53]
ChCl:methacrylic acid 1:2	Functional monomer for polymerization	Selective recognition and separation of bovine hemoglobin (BHb) protein	DES-MIPs showed a much higher adsorption capacity, rapid binding kinetics, and high imprinting factor for the BHb, compared with the magnetic DES-NIPs (NIP = nonimprinted polymers); Magnetic DES-MIPs presented highly recognition, specificity and selectivity	[64]
ChCl:ethylene glycol 1:3 ChCl:glycerol 1:3, 1:2, 1:4, 1:6 ChCl:1,4-butanediol 1:3	MIPs modifier: interaction with a functional monomer	Selective recognition and extraction of rutin, scoparone, and quercetin from Herba *Artemisiae Scopariae*	DES-MIPs of ChCl:glycerol 1:3 showed the best results, in comparison to other DES and MIPs	[54]
ChCl:ethylene glycol 1:2	Solvent	Recognition of clorprenaline and bambuterol in urine	The polymeric resins prepared in DES had higher adsorption capacity than the ones prepared in alcoholic solvents; 100% DES used as a polymerization solvent	[50]
ChCl:glycerol 1:2	MIPs modifier: interaction with a functional monomer	Purification of chloromycetin and thiamphenicol from milk	Higher extraction recoveries for DES-MIPs, than for MIPs and NIPs	[57]
ChCl:ethylene glycol 1:1 ChCl:glycerol 1:1 ChCl:propylene glycol 1:1	MIPs modifier: interaction with a functional monomer	Screening chloramphenicol in milk	Adsorption capacity of DES-MIPs better than MIPs; ChCl:EG-based DES-MIPs had the best adsorption ability and higher recoveries than MIPs and C18	[62]
Betain:ethylene glycol:water 1:2:1	MIPs modifier: interaction with a functional monomer	Adsorption of levoflaxacin or tetracycline from a millet extraction with a mixture of other antibiotics	The DES-MIPs showed better efficiency in recognition and specific adsorption than MIPs	[52]
Betaine:ethylene glycol:water 1:2:1	MIPs modifier: interaction with a copolymer	Recovery of levofloxacin from green bean extract, through SPE	DES-MIPs showed better adsorption capacity and higher recoveries of levofloxacin than MIPs, NIPs, Mesoporous Siliceous Material (MSM), DES-MSM, and C18	[58]
ChCl:ethylene glycol 1:2 ChCl:glycerol 1:2 ChCl:1,4-butanediol 1:2 ChCl:formic acid 1:2 ChCl:acetic acid 1:2 ChCl:propionic acid 1:2 ChCl:urea 1:2	MIPs modifier: interaction with a functional monomer	Purification of alkaloid isomers (theobromine and theophylline) from green tea	DES-MIPs of ChCl-urea 1:2 showed the best results, in comparison to other DES and ionic liquid modified-MIPs	[55]
Formic acid: methylltriphenylphosphonium bromide: chalcone (FA:Mtpp:Chal) 1:0.5:0.04, 1:0.5:0.05, 1:0.5:0.06,	Functional monomer and dummy template	Selective recognition of rutin and quercetin from molecular mixtures	1:0.5:0.05-based DES-MIP had the best adsorption capacity	[73]
Caffeic acid: ChCl:Formic acid (CA:ChCl:FA) 1:3:1.5, 1:4:2, 1:6:3	Functional monomer for polymerization	Adsorption of levofloxacin from millet extract	DES-MIPs of 1:3:1.5 CA:ChCl:FA more selective for detection and purification of levofloxacin	[65]
ChCl:ethylene glycol 1:3 ChCl:glycerol 1:3 ChCl:1,4-butanediol 1:3 ChCl:urea 1:3 ChCl:formic acid 1:3 ChCl:acetic acid 1:3 ChCl:propionic acid 1:2	MIPs modifier	Recognition of fucoidan and alginic acid from seaweed by magnetic solid-phase extraction	Best recovery using the ChCl:urea based DESs-magnetic MIPs The best DESs-magnetic MIPs was better than the respective MIPs and NIPs	[60]
ChCl: caffeic acid:ethylene glycol 1:0.1:1, 1:0.2: 1, 1: 0.3:1, 1:0.4:1	Template and functional monomer	Recognition of polyphenols	1:0.4:1-based DES-MIPs had the best adsorption capacity; DES-MIPs had better specific recognition and larger adsorption abilities than NIP, C18, and C8; Recognition of CA from polyphenol mixtures and in a real sample	[69]
ChCl:ethylene glycol 1:2 ChCl:glycerol 1:2 ChCl:1,4-butanediol 1:2 ChCl:urea 1:2 ChCl:formic acid 1:2 ChCl:acetic acid 1:2 ChCl:propionic acid 1:2	MIPs modifier	Purification of D-(+)-galactose, L-(−)-fucose, and D-(+)-mannose from seaweed, though SPE	Best recovery for the ChCl:urea-based DESs-Fe3O4@hybridMIPs; The best DESs-Fe3O4@HMIPs system was better than the respective Fe3O4@HMIPs, DES-HMIPs, and DES-NIPs	[59]
Allyltriethylammonium Chloride ([ATEAm]Cl):glycerol 1:1	Functional monomer for polymerization	Adsorption of lysozyme	DES-MIPs showed a good adsorption capacity, with a higher imprinting factor and higher specificity than other MIPs for lysozyme purification; 4 times recyclable	[66]
ChCl:urea 1:2 ChCl:ethylene glycol 1:2 ChCl:1,4-butanediol 1:2 ChCl:glycerol 1:2	MIPs modifier	Extraction of tanshinone I, IIA, and cryptotanshinone from *Salvia miltiorrhiza bunge*; glycitein, genistein, and daidzein from *Glycine max (Linn.) Merr*; and epicatechin, epigallocatechin gallate, and epicatechin gallate from green tea	Multiple template DES-MIPs reduced the experimental steps; The DES-MIPs tested were better than NIPs and MIPs, except for the ChCl-urea-based DES-MIP; Best extraction recoveries for ChCl-glycerol-based DES-MIP; DES-MIPs can be reused	[61]
ChCl:methacrylic acid (MAA) 1:2 Betaine/MAA/H_2_O 1:2:1	Functional monomer	Separation of (+)-catechin, (−)-epicatechin, and (−)-epigallocatechin gallate from black tea	Higher recoveries with DES-MIPs than MAA-MIPs or NIPs; The ChCl-MAA bases DES-MIP had slightly better results	[70]
ChCl:oxalic acid:ethylene glycol 1:1:1, 1:1:2, 1:1:3 ChCl:oxalic acid:glycerol 1:1:3 ChCl:oxalic acid:propylene glycol 1:1:1 ChCl:caffeic acid:ethylene glycol 1:1:1	Functional monomer	Selective recognition and separation of theophylline, theobromine, (+)-catechin hydrate, and caffeic acid from green tea	The ChCl:OA:PG based DES-MIPs has the best recovery results and was better than the respective DES-NIP, MIP, NIP, and MIPs made from conventional monomers (MAA and AM)	[68]
ChCl:acrylic acid 1:2	Additive functional monomer	Isolation of transferrin from human serum	Selective adsorption over protein mixtures	[71]
ChCl:formic acid 1:2 ChCl:acetic acid 1:2 ChCl:propionic acid 1:2 ChCl:urea 1:2	MIPs modifier: interaction with a functional monomer	Selective recognition and separation of Fucoidan and Laminarin	DES used for modification of MIPs by interaction with the functional monomer; DES-MIPs of ChCl-urea 1:2 showed the best results, in comparison to other DES, ionic liquid modified-MIPs and nonmodified MIPs	[56]
ChCl:DHBA:EG 1:1:1, 1:1:2, 1:1:3	Template and functional monomer	Extraction of 3,4-dihydroxybenzoic acid (DHBA)	DES-MIPs showed higher recoveries than MIPs, NIPs, and the corresponding DES-NIPs; 1:1:2 ChCl:DHBA:EG-based DES-MIPs showed the highest recoveries of 3,4-DHBA and better adsorption capacity, imprinted factor, and selectivity than the conventional functional monomer 4-vinylpyridine	[67]
(APTMACl):urea 1:2	Functional monomer for polymerization	Separation of bovine hemoglobin from a complex sample	DES-MIPs separated effectively BHb from calf blood; DES-MIPs could be recycled at least 3 times	[48]
ChCl:ethylene glycol 1:2, 1:3, 1:4	Binary green solvent and MIP modifier: porogen (mixture with ionic liquid)	Drug delivery of Fenbufen	The binary green system was the unique solvent used for all the polymerization reagents; It was also a good dispersant for the single-walled carbon nanotubes	[51]
ChCl:ethylene glycol 1:2	MIPs modifier: porogen	Determination of Levofloxacin in human plasma	DES-MIPs better than DES-NIPs; DES-MIPs efficiently applied to examine levofloxacin from human plasma of hospitalized patients	[63]
ChCl:acrylic acid 1:2	Functional monomer	Recognition and good antibacterial properties for β-lactoglobulin in milk	Surface DES-MIPs prepared to facilitate further template removal; The produced polymeric system presented good adsorption and selectivity for β-lactoglobulin, was reusable, and showed antibacterial activity	[72]

**Table 3 polymers-11-00912-t003:** Summary of some reported studies on polymer extraction with DES, from 2017 to 2019.

DES (molar ratio)	Polymer extracted	Raw material	Description	Ref.
ChCl:urea (U) 1:2 ChCl:ethylene glycol (EG) 1:2 ChCl:glycerol (GOH) 1:2 ChCl:lactic acid (LA) 1:2 ChCl:acetic acid (HAc) 1:2 ChCl:oxalic acid (OA) 1:1, 1:0.8, 1:0.6, 1:1.2	Collagen	Cod skin	Extraction abilities: ChCl:OA > ChCl:HAc > ChCl:La > ChCl:EG > ChCl:GOH > ChCl:U; Better extraction efficiency for 1:1 ChCl:oxalic acid; Extraction influenced by the DES viscosity, acidity, and free hydrogen protons	[95]
1:2 of ChCl with lactic acid, 1,4-butanediol, ethylene glycol, urea, 1,6-hexanediol, glycerol, oxalic acid, malonic acid, citric acid, malic acid, propylene glycol, L-(+)-tartaric acid, maleic anhydride, or thiourea	Chitin	Shrimp shells (*Marsupenaeus japonicas*)	Highest yield obtained for the DES of ChCl:oxalic acid, but the most selective (purest chitin) was ChCl:malonic acid DES; Higher yields than conventional extraction for all DES tested; Extracted chitin formed films with similar properties to films from commercial chitin	[94]
ChCl:oxalic acid 1:2	Keratin	Wool	Extraction assisted with dialysis High solubility of wool in the DES	[99]
ChCl:oxalic acid 1:2	Keratin	Rabbit Hair	Efficient dissolution and extraction of keratin from rabbit hair; Extraction assisted with dialysis	[100]
ChCl:malic acid 1:1	Chitin	Shrimp shells	Efficient extraction of chitin, demineralized and deproteinized	[101]
ChCl:lactic acid (LA) 1:2 ChCl:urea (UA) 1:2 ChCl:oxalic acid (OA) 1:1, 1:2 ChCl:potassium hydroxide 1:4	Lignin and cellulose	Wood sawdust of beech (*Fagus sylvatica*)	Oxalic acid and urea-based DES (acidic) were selective for lignin extraction, while ChCl:KOH (alkaline) was selective for extracting cellulose	[96]
ChCl:lactic acid 1:9	Lignin	Wood	Lignin 80% pure	[102]

**Table 4 polymers-11-00912-t004:** General characteristics of plasticizers and their influence in polymers properties.

Plasticizers characteristics	Influence of plasticizers in polymers
• Inert	• Decrease the melting or glass transition temperature (*T*_g_) of polymers
• Low molecular weight	• Preservation of the polymer elasticity
• Low vapor pressure	• Higher thermostability

**Table 5 polymers-11-00912-t005:** Resume of polymer plasticized using deep eutectic solvents and properties acquired by the polymers.

Polymer plasticized	DES	Properties conferred	Ref.
Starch	ChCl:imidazole 3:7, 2:3 Glycerol:imidazole 1:1, 3:7	Lower tendency to retrogradation Transparent and elastic films (thermoplasticized)	[118]
	Citric acid:imidazole 3:7 Malic acid:imidazole 3:7	Not suitable for starch plasticizing	
	ChCl:urea 1:2 ChCl:imidazole 3:7	Dependent of additive (consult article)	[119]
Chitosan films	ChCl:malic acid 1:1	Tailored ductility with DES content Lower T_g_ Good solubility in water	[114]
	ChCl:lactic acid 1:1	Transparent films Lower tensile strength and Young’s modulus (higher flexibility) Higher water vapor permeability (WVP), water solubility, and water sorption	[120]
	ChCl:urea 1:2	Enhanced film flexibility Reduced water uptake Improved ionic conductivity	[112]
	ChCl:malic acid 1:1, ChCl:lactic acid 1:1, ChCl:citric acid acid 1:1, ChCl:glycerol 1:2	Transparent films Elasticity, tensile strength, and WVP tuned by chitosan type and DES composition	[113]
Chitosan-carboxymethyl cellulose membrane	ChCl:urea 1:2	Higher thermal stability Improved flexibility	[121]
Agar films	ChCl:urea 1:2	Good mechanical resistance and improved elasticity in comparison to aqueous agar films	[85]
Cellulose films	ChCl:glycerol 1:2 ChCl:glucose 1:2 ChCl:urea 1:2	Highly improved ductility	[111]
	Tetrabutylammonium bromide:propylene carbonate 1:2 Tetrabutylammonium bromide:ethylene carbonate 1:2	Improved thermoformability	
Blend of starch and poly-ε-caprolactone (SPCL)	Glucose:citric acid 1:1 ChCl:sucrose 1:1, 4:1 ChCl:citric acid 1:1 ChCl:xylose 2:1, 3:1 Glucose:tartaric acid 1:1 Citric acid:sucrose 1:1	Lower Young’s modulus and ductility Enhancement of supercritical foaming	[116]
*Momordica charantia* bioactive polysaccharide	ChCl:glycerol 1.5:1, 1:1, 1:1.5, 1:2, 1:3	Improved flexibility (higher tensile stress and Young’s modulus) Higher thermal stability Higher water adsorption and WVP Antioxidant and antimicrobial activity	[117]

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
