# Peer review of "Polymer Science and Engineering Using Deep Eutectic Solvents"

_polymers, 2019, doi:10.3390/polym11050912_

Reviewer 1 Report

The present contribution by Duarte and co-workers nicely reviews the use of Deep Eutectic Solvents in polymeric chemistry by giving a clear and general overview of this topic (since 2016) ranging from the use of DESs as monomers or modification agents to their employment for the formulation of materials. In general, the review is well-written in an easy-to-follow manner. Thus, I can recommend the acceptance of this review after the following minor revisions:

1) In the first line of the introduction (line number 24) the authors nicely presented the coinage of the term Deep Eutectic Solvent by Abbott. However to be totally precise, the selected reference (number 1) describes the synthesis of eutectic mixtures containing choline chloride/Urea or choline chloride/amines and not those containing metallic salts.

2) Introduction, page 1 line 37. It would be ideal if the authors could also introduce the reference by Anastas and Warner in which the principles of Green Chemistry were firstly reported (Anastas, P. T.; Warner, J. C. Green Chemistry: Theory and Practice, Oxford University Press: New York, 1998, p.30.)

3) Page 1, line 41. The authors only use one reference to cover all the applications of DES. However, in the modest opinion of this reviewer other important reviews are missing. For example, the authors could use the following ones: i) Eur. J. Inorg. Chem., 2015, 5147; ii) Eur. J. Org. Chem. 2016, 612.

4) Page 2, line 62. At the end of this sentence the authors should quote that 2 previous reviews in this field have nicely covered this topic. Although these two reviews appear in the reference section (reference 8 and reference 35) they should also be commented here.

5) page 3, line 74. The authors deeply commented here the importance of H-bonds network in DES. This network has been already study in the liquid state of the mixture choline chloride/Urea by neutron diffraction and atomistic modelling (Green Chem., 2016, 18, 2736).

6) Table 2 and 3 are far too big. Is it possible to reduce their length (more than 2 pages in the case of table 2)??

7) Finally, a recent paper that opens the door to the synthesis of polystyrenes by employing organolithium reagents (anionic polymerization) in DES is missing. Please add and comment the following reference: ChemSusChem, 2019, DOI:10.1002/cssc.201900533

Thus, this is a nice and timing compilation of the recent developments of polymeric chemistry in DESs that really deserves publication in Polymers (after revision of the aforementioned points).

Author Response

Point 1: In the first line of the introduction (line number 24) the authors nicely presented the coinage of the term Deep Eutectic Solvent by Abbott. However to be totally precise, the selected reference (number 1) describes the synthesis of eutectic mixtures containing choline chloride/Urea or choline chloride/amines and not those containing metallic salts.
Response 1: Thank you so much for the constructive critical revision.
It is true that the article is about eutectic mixtures containing choline chloride/Urea or choline chloride/amines. However, it is also true that this was the first time that Abbott referred the term ‘deep eutectic’ in respect to previously reported metal salts-based mixtures which he considered to fulfill the DES definition, as cited here “In the extreme, ambient temperature molten salts have been formed by mixing quaternary ammonium salts with metal salts. This type of ionic liquid can be viewed as a deep eutectic…”.

Point 2: Introduction, page 1 line 37. It would be ideal if the authors could also introduce the reference by Anastas and Warner in which the principles of Green Chemistry were firstly reported (Anastas, P. T.; Warner, J. C. Green Chemistry: Theory and Practice, Oxford University Press: New York, 1998, p.30.)
Response 2: The reference was added in line 38 (ref 12).

Point 3: Page 1, line 41. The authors only use one reference to cover all the applications of DES. However, in the modest opinion of this reviewer other important reviews are missing. For example, the authors could use the following ones: i) Eur. J. Inorg. Chem., 2015, 5147; ii) Eur. J. Org. Chem. 2016, 612.
Response 3: The suggested references were included (line 43, ref 16 and 17)

Point 4: Page 2, line 62. At the end of this sentence the authors should quote that 2 previous reviews in this field have nicely covered this topic. Although these two reviews appear in the reference section (reference 8 and reference 35) they should also be commented here.
Response 4: We totally agree that those references nicely address the DES features and its advantages to the discussed applications: polymerization in ref 8 and synthesis in ref 35. However, we think that these reviews do not address fundamental studies of DES and polymers as this would require to study their physicochemical properties, as we address in the examples described in that paragraph, for example, molecular solvation or thermodynamic phase equilibria.

Point 5: page 3, line 74. The authors deeply commented here the importance of H-bonds network in DES. This network has been already study in the liquid state of the mixture choline chloride/Urea by neutron diffraction and atomistic modelling (Green Chem., 2016, 18, 2736).
Response 5: The suggested reference was added and commented as follows: “Fundamental data on the influence of the DES hydrogen networking in its 3D structure can be acquired to predict its solvation capacity and used as a pre-selection parameter for certain applications. An example is the work of Hammond et al. on neutron diffraction and atomistic modelling to acquire the probable liquid structure of choline chloride-urea DES [21]. Considering DES-polymer systems, this type of preliminary information for several DES systems and/or different component ratios, could be used to select or adjust suitable DES for polymer solvation. Still, the DES-polymer systems are more complex than DES in its isolated form, requiring the study of the combined system to have more accurate information on their interaction.” (line 76-83)

Point 6: Table 2 and 3 are far too big. Is it possible to reduce their length (more than 2 pages in the case of table 2)??
Response 6: The authors agree that the tables are large, however, the information included is in not available in the text, and we tried to summarize the findings in a schematic and accessible way to the reader. For these reasons, we believe that it is valuable to include them as they are.

Point 7: Finally, a recent paper that opens the door to the synthesis of polystyrenes by employing organolithium reagents (anionic polymerization) in DES is missing. Please add and comment the following reference: ChemSusChem, 2019, DOI:10.1002/cssc.201900533
Response 7: The suggested reference were added and commented as follows (lines 259-264): “Additionally, a more recent work published the use of ChCl.glycerol 1:2 molar ratio DES as reaction media for oleofin anionic polymerization [82]. The authors highlighted the advantages of the protic and polar character of DES for the polymerization and reported that the selective use of glycerol as hydrogen bond donor instead of urea, lactic acid or oxalic acid increased the polymerization performance. These examples emphasize, once again the potential of DES versatility to be tailored for specific applications.”
Thank you for the collaboration in enriching this review.

Reviewer 2 Report

The manuscript "Breakthroughs in polymer science using deep eutectic solvents" offers an interesting review of recent literature featuring deep eutectic solvents (DES) and polymers. The manuscript is, for the most part, well-written and covers most of the key recent literature. The manuscript could be published in Polymer subject to a few suggestions.

The title of the manuscript is not accurate. There have been no substantive breakthroughs in polymer science made due to the use of DES'. I agree that the use to DES as solvents for reactions and even as chemical components of reactions has increased, but this has not resulted in fundamental or important "breakthroughs" of any note for the wider chemistry or scientific community.

Line 31 - when commenting on the mechanism of DES formation the authors omit the large body of computational chemistry work in the field that has gone a long way in determining the key interactions in DESs. See for example the works of the Kirchner group, Page group or Hunt group among others. I understand the focus of this review is the use of DES' with polymers, but the authors cannot comment that little is known of the mechanism of DES formation when this is not the case.

Line 34 - "The exponential environment sensibilization" must be reworded. I understand the authors are trying to say that society is more increasingly concerned about the environment, but this particular phrase does not make sense.

Line 59 - "personalized optimization" needs to be reworded.

The authors suggest multiple times that results from literature signify that DES may the "tailored" for a specific purpose. It would thus greatly increase the impact of this review if the authors could provide a summary a key DES properties, such as hydrogen bond capacity, and their impact/relevance for a particular application. In this way the reader might use this manuscript as a "toolkit" for developing new DES' or selecting a DES for a particular application.

Author Response

Point 1: The title of the manuscript is not accurate. There have been no substantive breakthroughs in polymer science made due to the use of DES'. I agree that the use to DES as solvents for reactions and even as chemical components of reactions has increased, but this has not resulted in fundamental or important "breakthroughs" of any note for the wider chemistry or scientific community.
Response 1: We changed the title to “Polymer science and engineering using deep eutectic solvents”

Point 2: Line 31 - when commenting on the mechanism of DES formation the authors omit the large body of computational chemistry work in the field that has gone a long way in determining the key interactions in DESs. See for example the works of the Kirchner group, Page group or Hunt group among others. I understand the focus of this review is the use of DES' with polymers, but the authors cannot comment that little is known of the mechanism of DES formation when this is not the case.
Response 2: Thank you for pointing out the important contribution of the mentioned groups. We included some of their work to support the mentioned mechanisms most probably responsible for DES formation. However, the computational chemistry works developed are normally referring to specific DES system (commonly, ChCl-based DES). Even though these studies are highly important as a starting point, they might not be extensible to other DES systems. It is also important to highlight that the mechanisms reported are hypothesis, as exemplified in the following citations: “The current consensus is that Type III DES form via the intercalation of the salt lattice by the HBD...” (Phys.Chem.Chem.Phys.,2017, 19, 3297); “Thus it may be reasonable to suggest an alternative picture to the formation of a [Cl(urea)2] complexed anion leading to a reduced cation–anion interaction.” (Phys.Chem.Chem.Phys.,2016, 18, 18145). Moreover, some questions are also raised: “Thus, it seems questionable if charge delocalization occurring through hydrogen bonding between the halide anion and the organic compound is responsible for the deep eutectic melting point” (ChemPhysChem 2016, 17, 3354 – 3358)
So, we used the suggested work as references to support that the DES formation “probably occurs through hydrogen bonding, electrostatic interactions and/or Van der Waals interactions”, but we still maintained the idea that ‘The mechanism of DES formation is not yet well understood’, as more extensive studies need to be made to create a solid baseline knowledge on the field.

Point 3: Line 34 - "The exponential environment sensibilization" must be reworded. I understand the authors are trying to say that society is more increasingly concerned about the environment, but this particular phrase does not make sense.
Response 3: The referred expression was reformulated as follows: “The exponential concern with environment issues highly motivated the development of green technologies.”

Point 4: Line 59 - "personalized optimization" needs to be reworded.
Response 4: In the context, the expression was altered, as follows: “The fundamental knowledge of their properties for those applications is, hence, crucial for their optimization and development.”

Point 5: The authors suggest multiple times that results from literature signify that DES may the "tailored" for a specific purpose. It would thus greatly increase the impact of this review if the authors could provide a summary a key DES properties, such as hydrogen bond capacity, and their impact/relevance for a particular application. In this way the reader might use this manuscript as a "toolkit" for developing new DES' or selecting a DES for a particular application.
Response 5: Thank you for the constructive suggestion. In this review we focused on the different applications of DES in polymer science. The number of studies reported in the literature regarding the same exact topic is not yet significant to extract the information required to design a toolkit, as the reviewer suggests. This would in fact be a great achievement for the DES community, unfortunately this is not yet possible.
However, we already had some examples that can be used as a basis for that: “The hydrogen bond basicity of the DES used showed to be preeminent for the effective polymerization of that polymer [78], highlighting once again the potential of DES versatility to be tailored for specific applications.”
Still, we added a general example of how to apply DES characterization on their choice for solvation applications: “Fundamental data on the influence of the DES hydrogen networking in its 3D structure can be acquired to predict its solvation capacity and used as a pre-selection parameter for certain applications. An example is the work of Hammond et al. on neutron diffraction and atomistic modelling to acquire the probable liquid structure of choline chloride-urea DES [21]. Considering DES-polymer systems, this type of preliminary information for several DES systems and/or different component ratios, could be used to select or adjust suitable DES for polymer solvation. Still, the DES-polymer systems are more complex than DES in its isolated form, requiring the study of the combined system to have more accurate information on their interaction.” (line 76-83)
Moreover, we added a more particular example: “Additionally, a more recent work published the use of ChCl.glycerol 1:2 molar ratio DES as reaction media for oleofin anionic polymerization [82]. The authors highlighted the advantages of the protic and polar character of DES for the polymerization and reported that the selective use of glycerol as hydrogen bond donor instead of urea, lactic acid or oxalic acid increased the polymerization performance. These examples emphasize, once again the potential of DES versatility to be tailored for specific applications.” (lines 259-264)
Thank you for your contribution in the improvement of this review.

Round  2

Reviewer 2 Report

I am satisfied the authors have responded to the points raised in the initial review adequately.

Author Response

Thank you